# Design and Simulation of Au/SiO_2_ Nanospheres Based on SPR Refractive Index Sensor

**DOI:** 10.3390/s23063163

**Published:** 2023-03-16

**Authors:** Meng Sun, Yutong Song, Haoyu Wu, Qi Wang

**Affiliations:** College of Sciences, Northeastern University, Shenyang 110819, China

**Keywords:** surface plasmon resonance, Au/SiO_2_ nanospheres, Au/SiO_2_ nanorods

## Abstract

In this paper, three different structures of surface plasmon resonance (SPR) sensors based on the Kretschmann configuration: Au/SiO_2_ thin film structure, Au/SiO_2_ nanospheres and Au/SiO_2_ nanorods are designed by adding three different forms of SiO_2_ materials behind the gold film of conventional Au-based SPR sensors. The effects of SiO_2_ shapes on the SPR sensor are investigated through modeling and simulation with the refractive index of the media to be measured ranging from 1.330 to 1.365. The results show that the sensitivity of Au/SiO_2_ nanospheres could be as high as 2875.4 nm/RIU, which is 25.96% higher than that of the sensor with a gold array. More interestingly, the increase in sensor sensitivity is attributed to the change in SiO_2_ material morphology. Therefore, this paper mainly explores the influence of the shape of the sensor-sensitizing material on the performance of the sensor.

## 1. Introduction

Surface plasmon resonance (SPR) sensors are sensitive to changes in the refractive index of the medium in contact with the metal film, and when the refractive index of the medium changes, the resonance wavelength or angle will be changed, thus realizing the detection of a specific medium [1,2,3]. SPR sensors are widely used in food safety [4,5], biology [6,7,8], and gas and temperature monitoring [9,10,11,12] because of their label-free detection, low cost, real-time measurement, and high sensitivity. SPR sensors include prism coupling [13,14,15], waveguide coupling [16], fiber coupling [17,18,19], and grating coupling [20].

The Kretschmann structure [21] is a common SPR sensor structure consisting of a prism, a metal film, and a medium [22]. A polarized light illuminates the prism and undergoes attenuated total internal reflection at the metal/prism interface. The evanescent wave resonates with the plasmon propagating on the metal medium’s surface, causing reflected light absorption [23]. The electric field intensity is strongest at the intersection of the metal and the medium, and decreases exponentially with the increase of medium depth.

In recent years, SPR sensors have developed very rapidly. However, when detecting media with low concentration or small molecules, SPR sensors would have low sensitivity due to the small range of refractive index changes that these media bring about. Therefore, various strategies have been developed to improve the sensitivity of SPR sensors, including metallic nanowires [24,25], nanohole arrays [26,27], metal oxides [15,28], nanoparticles and so on. Although gold has good chemical stability, the surface of the gold film is too smooth to adsorb a large number of molecules, thus limiting the sensitivity of the sensors.

In 2004, Brolo et al. demonstrated the feasibility of using periodic metal nanostructures as sensors alone [29]. A 200 nm diameter periodic hole array was fabricated on a 100 nm gold film. The sensing sensitivity of the periodic nanopore chip was about 400 nm/RIU. Then, Pang et al. increased the sensitivity of the nanopore array spectrum detection to 1520 nm/RIU [28]. In 2020, Armin Agharazy Dormeny et al. made SPR sensors with different gold structures arranged periodically, and explored the influence of sunken or raised gold nanorods, and nanorods on the performance of SPR sensors [24,30].

Recent advances in nanotechnology have led to the emergence of many different types of SiO_2_ structures with a variety of heights, shapes and sizes, mostly used in biosensors [31]. SiO_2_ has a high yield, low cost, good chemical stability, strong film, low refractive index, and unique optical properties. It has a variable bandwidth, small particle size, large specific surface area and large volume voids, and can adsorb more molecules of the sensing medium. Therefore, it is often used as a composite material to improve the reactants’ reaction efficiency, stability and adsorption capacity [32,33]. The above sensor discusses the influence of the form of the sensitizing material Au on the performance of the SPR sensor. Based on the above work and the advantages of SiO_2_ material, this paper studies the influence of the form of SiO_2_ material on the performance of the SPR sensor.

In this work, the SPR sensors are simulated by combining silica material with gold using the fluctuating optics module of COMSOL Multiphysics software [34]. The purpose of this paper is to explore the effect of three different forms of SiO_2_ materials on the performance of SPR sensors. Thus, sensors with Au/SiO_2_ thin film structure, Au/SiO_2_ nano-spheres and Au/SiO_2_ nanorods are designed and their sensitivity, as well as *FOM*, are analyzed and compared, respectively. Following the optimization of their structural parameters, it is found that the Au/SiO_2_ nanospheres had the best sensitivity and *FOM*, and could significantly enhance the performance of the SPR sensors. This indicates that the device with the Au/SiO_2_ nanosphere structure not only enhances the performance of SPR sensors, but also has important implications for the measurement of SPR sensors.

## 2. Theoretical Analysis and Device Modeling and Simulation Setup

The necessary condition for exciting surface plasmon (SP) is that the wave vector of polarized incident light ($K_{c}$) should be equal to the wave vector of surface plasmon ($K_{sp}$) [35]:(1)$$k_{sp}=k_{c}$$

The wave vector of surface plasmon ($K_{sp}$) and the wave vector of polarized incident light ($K_{c}$) can be expressed as follows [36,37]:(2)$$K_{sp}=\frac{\omega_{c}}{c}\sqrt{\frac{\varepsilon_{m}\varepsilon_{d}}{\varepsilon_{m}+\varepsilon_{d}}}$$
(3)$$K_{c}=n_{p}k_{0}sin\theta$$
where $\varepsilon_{m}$ represents the RI of the metal film, $\varepsilon_{d}$ represents the RI of the sensing medium, $n_{p}$ is the RI of the prism, λ is the incident wavelength and θ is the incident angle.

The wavelength corresponding to the minimum reflectance is the resonance wavelength. The change in resonance wavelength is caused by the change in the refractive index of the sensing medium, and the sensitivity can be defined as [38]:(4)$$S=\frac{∆\lambda_{SPR}}{∆n}$$
where $\Delta_{SPR}$ is the change of SPR resonance wavelength, and ∆n is the change of refractive index.

The full width at half maxima (FWHM) of the SPR reflectance spectrum and Figure of merit (*FOM*) are also important factors in evaluating the performance of a sensor, and the *FOM* can be calculated as follows [5]:(5)$$FOM=\frac{S}{FWHM}$$
where FWHM is the geometric parameter of the SPR curve. While paying attention to the sensitivity of the sensor, it is also necessary to pay attention to its FWHM. *FOM* can comprehensively consider the sensitivity and FWHM. Therefore, it is expected that the sensor can have greater sensitivity and smaller FWHM, that is, the sensor with a larger *FOM* will have better performance.

The electromagnetic module in COMSOL is mainly calculated according to the Max-well equation. In this paper, three different structures of sensors are simulated with the prism material BK7. The refractive index of BK7 prism varies with the wavelength by the following formula [39]:(6)$$n^{2}=1+\frac{1.03961212\lambda^{2}}{\lambda^{2}-0.00600069867}+\frac{0.231792344\lambda^{2}}{\lambda^{2}-0.0200179144}+\frac{1.01046945\lambda^{2}}{\lambda^{2}-103.560653}$$

The alcohol is selected as a sensing medium and the refractive index of the alcoholic solution varies from 1.330 to 1.365. Figure 1 shows the diagram of the SPR sensor. The initial setting of the key parameters are as follows: incidence angle (θ) = 74 deg, incident light wavelength = 632.8 nm and the thickness of the old film = 50 nm. The Lorentz–Drude model in the experimental section of the COMSOL material library is adopted for Au and subsequent SiO_2_ materials.

In this paper, the performance of the SPR sensors with three different structures: Au/SiO_2_ thin film structure, Au/SiO_2_ nanospheres structure and Au/SiO_2_ nanorods structure are investigated by using the wavelength modulation method.

## 3. Results and Discussion

In this paper, two-dimensional and three-dimensional modeling and simulation studies on SPR sensors with the Kretschmann structure are carried out using COMSOL software [40]. Based on traditional SPR sensors with pure Au film, three kinds of SPR sensors with different structures, i.e., Au/SiO_2_ thin film structure, Au/SiO_2_ nanospheres structure and Au/SiO_2_ nanorods structure are designed, and the specific structure diagram is shown in Figure 1. The performance of the SPR sensors with the three different SiO_2_ form structures is investigated.

### 3.1. Au Structure

The SPR sensor has the lowest reflectance and the best sensing effect when a Au film of 50 nm is deposited on a glass prism (BK7). The Kretschmann gold film sensor is simulated by COMSOL, and five different thicknesses of the gold film (40 nm, 45 nm, 50 nm, 55 nm, 60 nm) are added behind the prism. Figure 2a shows the SPR curves of gold films of different thicknesses when the fixed measuring medium is water (*n* = 1.33). From Figure 2a, it is found that the variation of resonance wavelength of the gold film with different thicknesses is concentrated in the range of 630–650 nm. The resonance wavelength increases with the increase in thickness. A good sensor requires low reflectance and high-quality factor, so it can be found from Figure 2b that the 50 nm gold film just meets the above requirements. In addition, the 50 nm gold film is widely used in a variety of scientific research articles, has good comparability and is basic. Follow-up studies are carried out on this basis.

Thus, the SPR sensor with a pure 50 nm gold structure is simulated using COMSOL, as shown in Figure 1a. As can be seen from Figure 3a, when the refractive index of the medium increases from 1.330 to 1.365, the SPR curve shifts to the right, the resonance wavelength shows a red-shift shape, and the wavelength shift of SPR is 77 nm. Figure 3b shows that there is a good linear relationship between the resonance wavelength and the refractive index with the correlation coefficient R^2^ = 0.9916. The higher the degree of linear fitting, the smaller the relative error. The relation between resonant wavelength and refractive index is better and the accuracy of the measurement of the SPR sensor is higher. Its sensitivity can reach 2192.62 nm/RIU.

### 3.2. Au/SiO_2_ Thin Film Structure

In order to obtain the excitation effect of the gold film and the enhancement effect of SiO_2_ at the same time, the SiO_2_ film is added behind the gold film to construct the SPR sensor with a Au/SiO_2_ film structure without changing the above parameters, as shown in Figure 1b. Under the condition that the thickness of the gold film remains unchanged, the thickness of the SiO_2_ film was optimized by calculating the performance of the sensors with different SiO_2_ layer thicknesses in the range of 10–90 nm (10 nm for each increase). The simulation results are shown in Figure 4.

From Figure 4, it can be observed that under different SiO_2_ film thicknesses, the sensitivity and *FOM* both increase first and then decrease. Considering the sensitivity and *FOM*, we can find that when the thickness of the SiO_2_ film is 20 nm, the sensitivity and *FOM* of the SPR sensor are the highest, and the SPR sensor has the best performance. The sensor with 50 nm Au and 20 nm SiO_2_ film composite is selected as the sensor with the best performance.

Then, the sensor is used to test the media with different refractive indices ranging from 1.330–1.365. According to Figure 5a, the SPR wavelength shifted 77.9 nm, which is red-shifted for 0.9 nm compared with that of the gold structure. Figure 5b shows the relationship between the refractive index and resonance wavelength and their linear fitting. The refractive index is linearly correlated with the resonance wavelength with the correlation coefficient R^2^ = 0.9934. The sensitivity can reach 2230.2 nm/RIU, which is 1.71% higher than that of the previous gold structure, but the improvement is insignificant. The increased sensitivity may be due to the SiO_2_ material’s inherent refractive index. When light enters the surface of the gold film from the prism and causes surface plasmon resonance in SiO_2_ material, the corresponding resonance wavelength will be redshifted and increase its sensitivity.

### 3.3. Au Array Structure

A gold pore array with the same period and size as the SiO_2_ nanospheres in this work is established by COMSOL in Figure 6, which is more convenient for the sensitivity comparison. (Distance = 10 nm, Radius = 20 nm).

As can be seen from Figure 7a, when the refractive index of the medium increases from 1.330 to 1.365, the SPR curve shifts to the right, the resonance wavelength shows a redshifted shape, and the wavelength shift of SPR is 82 nm. Figure 7b shows that there is a good linear relationship between the resonance wavelength and the refractive index with the correlation coefficient R^2^ = 0.9909. Its sensitivity can reach 2282.8 nm/RIU. Compared with the gold film, the sensitivity of the gold array is improved by 4.11%. The simulation results show that the void structure improves the sensitivity of the sensor based on the pure gold film. Follow up studies are carried out on this basis.

### 3.4. Au/SiO_2_ Nanospheres

To further improve the sensitivity and *FOM*, and change the shape of the SiO_2_ mate-rial, periodic SiO_2_ nanospheres were embedded in a 50 nm gold film to form an SPR sensor with a periodic Au/SiO_2_ nanosphere structure, as shown in Figure 1c. According to the literature, it can be found that a linear relationship between period and size and FWHM and resonance wavelength [41]. Therefore, the study period and size are of great significance to this paper. However, the period of the device is the sum of the distance and ×radius in this work. In order to accurately consider the changes of multiple periods in the following articles, we divide the period into two variables: radius and distance. Therefore, two factors need to be considered, i.e., the radius of the nanospheres and the distance between the nanospheres.

Under the condition that the thickness of the gold film remains unchanged, in order to obtain the best combination of the radius of the SiO_2_ nanosphere and the distance between them, the sensing performance of the sensor with different combinations of SiO_2_ nanosphere radius, in the range of 10–30 nm (increasing 5 nm each time), and the distance between spheres ranging from 10–40 nm (increasing 5 nm each time) is calculated, and the simulation results are shown in Figure 8. It is observed that in the transverse direction, the sensitivity increases with the increase of the nanosphere radius, but *FOM* increases first and then decreases. It is observed in the longitudinal direction that the sensitivity decreases with the increase of the distance between nanospheres.

Considering the performance of both sensitivity and *FOM*, a set of optimal solutions is selected from Figure 8a–d, respectively, and the four sets of data were compared in detail in Table 1. By observing Table 1, it can be seen that the SPR sensor has the highest sensitivity and *FOM* when the spacing between the SiO_2_ nanospheres is 10 nm and the radius is 20 nm. At this time, the SPR sensing performance is optimal, thus making it the best choice.

The device is used to test media with different refractive indices ranging from 1.330–1.365. Figure 9a shows that the SPR wavelength is shifted by 100 nm, which is redshifted by 23 nm compared with that of the gold structure. Figure 9b shows the relationship between the refractive index and resonance wavelength, and the corresponding linear fitting. The refractive index is linearly correlated with the resonance wavelength with the correlation coefficient R^2^ = 0.9923. The sensitivity of the sensors with this structure can reach 2875.4 nm/RIU. The sensitivity is significantly improved compared with the previous two structures (31.14% higher than sensors with a pure gold film structure and 28.93% higher than sensors with the Au/SiO_2_ thin film structure). At the same time, the *FOM* of this structure is also higher than that of the previous two structures. Compared with a gold film, the specific surface area of the SiO_2_ nanospheres will be increased, which can provide more contact sites with alcohol molecules. When adsorbing more alcohol molecules, the resonance wavelength will be redshifted. Thus, the Au/SiO_2_ nanospheres structure sensor improves the sensitivity.

Comparing a pore structure with a SiO_2_ nanoarray showed that while the structure increases the sensitivity above the bare film, the SiO_2_ nanosphere further increases sensitivity. In order to further understand how the SiO_2_ nanospheres improve the sensitivity of the sensor, the electric field distribution diagram of the structure is drawn in Figure 10.

Figure 10 shows the electric field distribution at the resonance wavelength for a dielectric refractive index of 1.330 in the Au/SiO_2_ nanosphere structure. The electric field intensity is close to the maximum when the reflectance is minimal. It can be seen from Figure 10 that the maximum electric field intensity is obtained at the interface between Au and SiO_2_ nanospheres, and then the electric field intensity decays continuously in the induced medium. The electric field distribution indicates that the reduced reflectivity is caused by the SPR phenomenon.

### 3.5. Au/SiO_2_ Nanorod Structure

To investigate whether the SiO_2_ nanorods affect the effect of SPR sensing, the SiO_2_ nanospheres are replaced with SiO_2_ nanorods. The Au/SiO_2_ nanorod structure of the SPR sensor is formed by inlaying SiO_2_ nanorods in a gold film, as shown in Figure 1d. In this structure, three factors need to be considered, namely the radius and height of the nanorods and the distance between the nanorods. The effects of the above three factors on the sensing effect are shown in Table 2, Table 3 and Table 4.

To further investigate the performance of the sensor, we varied the height, radius and spacing of the SiO_2_ nanorods. The geometric parameters of the SiO_2_ nanorods are initially set at 40 nm in height, 20 nm in radius and 20 nm in spacing. The control variable method was subsequently used.

First, the effect of the distance between the SiO_2_ nanorods on the sensing performance was investigated, as shown in Table 2. While keeping the radius and height constant, the sensitivity of the sensor decreases as the distance increases from 10 nm to 30 nm and the *FOM* increases first and then decreases. Taking both into account, it is found that the sensitivity and *FOM* of the sensor are relatively high and the SPR is the best when the distance is 20 nm.

Next, the effects of the radius of the SiO_2_ nanorods and the rod spacing on the sensitivity and *FOM* of the sensor were investigated. It is observed from Table 3 that both the sensitivity and *FOM* of the sensor increase first and then decrease as the radius increases from 10 nm to 35 nm. When the radius of the SiO_2_ nanorods is 20 nm, the sensitivity and *FOM* are high. It is observed in Table 4 that when the rod height increases from 20 nm to 60 nm, the sensitivity of the sensor increases, while the *FOM* decreases. The sensitivity and *FOM* are relatively maximum when the spacing between the SiO_2_ nanorods is 40 nm. Finally, the geometric parameters of the optimized SiO_2_ nanorods are: a height of 40 nm, a radius of 20 nm and a distance of 20 nm.

The device was used to test media with different refractive indices ranging from 1.330–1.365. Figure 11a shows that the SPR wavelength is shifted by 101 nm, which is red-shifted by 24 nm compared with the gold structure. Figure 11b shows the relationship between the refractive index, resonance wavelength and the corresponding fitting line. The refractive index is linearly correlated with the resonance wavelength with the correlation coefficient R^2^ = 0.9879. The sensitivity can reach 2902.5 nm/RIU. The sensitivity is improved by 32.38% compared with that of the pure gold film sensors, but the improvement is not significant compared with that of the sensors with a Au/SiO_2_ nanosphere structure. Nonetheless, the sensitivity of this structure is significantly higher than that of the first two structures.

The sensing performance of the SPR sensor with four different structures: Au array structure, Au/SiO_2_ thin film structure, Au/SiO_2_ nanospheres and Au/SiO_2_ nanorods were identified by comparing the slopes of the four fitted straight lines, as shown in Figure 12. The sensitivity of the SPR sensor with the Au/SiO_2_ nanosphere structure is 25.96% higher than that of the SPR sensor with the Au array structure. Although its sensitivity is lower than that of the sensor with the Au/SiO_2_ nanorods, its *FOM* is better. In summary, the SPR sensor with the Au/SiO_2_ nanosphere structure has higher sensitivity and good *FOM*, and there is a good linear correlation between the resonance wavelength and refractive index. In addition, the sensor with this structure has a wide measurement range and can measure media with a refractive index ranging from 1.330 to 1.365. The comparison results also show that the sensor sensitivity is greatly affected by the morphology of SiO_2_ material.

The device is compared with other sensors, and the results are shown in Table 5. A 200 nm diameter periodic hole array was fabricated on a 100 nm gold film. The sensing sensitivity of the periodic nanopore chip was about 400 nm/RIU [30]. The device was constructed from a gold film with Al_2_O_3_ grating, which had a sensitivity of 461.53 nm/RIU [42]. The above two articles show that the periodic arrangement of the nanomaterials has an impact on the performance of the sensor. The sensitivity of this paper is higher than that of these two papers, which indicates that the periodic arrangement of the materials and the change of morphology have an impact on the performance of the sensor. Although these results indicate that the SPR sensor designed in this work has a good performance in alcohol detection, it still needs to be further explored in the experiment. The experimental scheme of silicon dioxide synthesis can be referred to the literature [43,44].

## 4. Conclusions

In this paper, three different structures of SPR sensors based on the Kretschmann con-figuration: Au/SiO_2_ thin film structure, Au/SiO_2_ nanospheres and Au/SiO_2_ nanorods were designed, mainly by depositing a 50 nm gold film on the glass and adding three different forms of SiO_2_ materials behind the gold film. Through modeling and simulation, the effects of these three structures on the sensitivity and *FOM* of the sensors were explored and compared. It was found that the sensitivity of the SPR sensor could reach 2875.4 nm/RIU when the SiO_2_ nanospheres (with a radius of 20 nm and spacing between nanospheres of 10 nm) were inlaid in the 50 nm gold film. The sensitivity of the sensor with the Au/SiO_2_ nanosphere structure is 25.96% higher than that of the SPR sensor with a gold array structure. Compared with the sensor with the Au/SiO_2_ nanorod structure, the sensor with the Au/SiO_2_ nanosphere structure has a slightly lower sensitivity but enjoys a higher *FOM*. Therefore, SiO_2_ nanospheres have more influence on the sensitivity of the sensor, which just proves the influence of the shape of the sensitizing material on the sensitivity. The device is used to measure the resonant wavelengths of substances with different refractive indices. Based on the curve of resonant wavelength varying with the refractive index, it is found that the device has linear properties and can be used to identify substances with an unknown refractive index. The results of this device show that the different forms of SiO_2_ arranged theoretically.

## Figures and Tables

**Figure 1 sensors-23-03163-f001:**
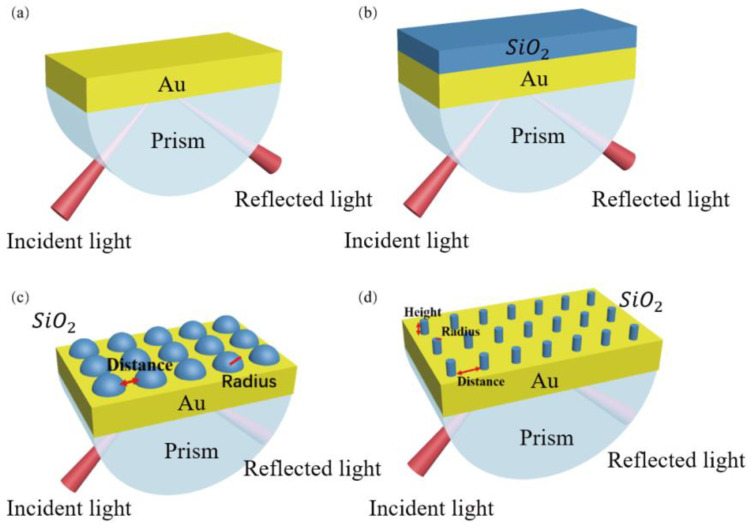
Schematic diagram of various structures of SPR sensors (**a**) Au film, (**b**) Au/SiO_2_ thin film, (**c**) Au/SiO_2_ nanospheres and (**d**) Au/SiO_2_ nanorods.

**Figure 2 sensors-23-03163-f002:**
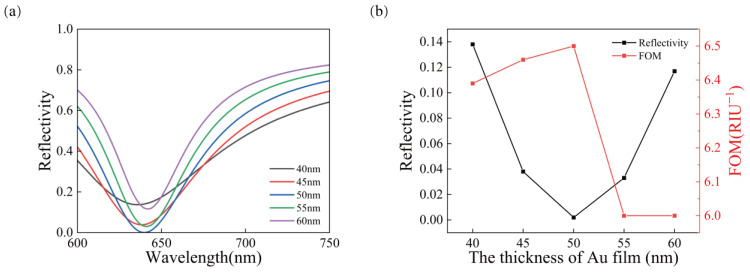
(**a**) SPR curves for the different thicknesses of gold film, (**b**) comparison of reflectivity and *FOM* at different gold film thicknesses.

**Figure 3 sensors-23-03163-f003:**
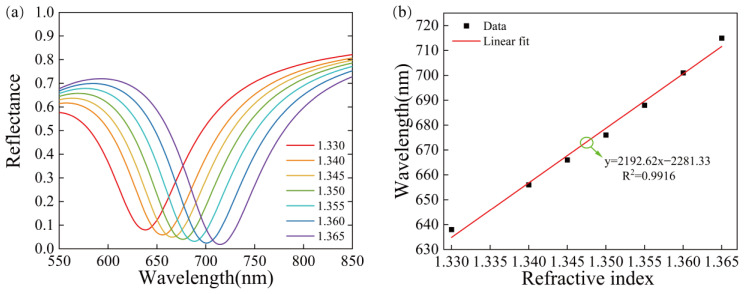
The reflectance curves of the Au structure, (**a**) SPR curves for different refractive indices of the pure gold film structure, (**b**) relationship between the refractive index and the resonance wavelength and linear fitting.

**Figure 4 sensors-23-03163-f004:**
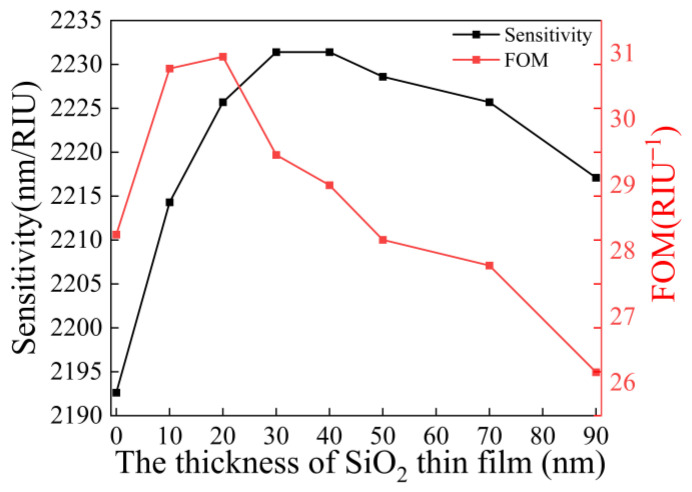
Comparison of the sensitivity and *FOM* at different silica film thicknesses.

**Figure 5 sensors-23-03163-f005:**
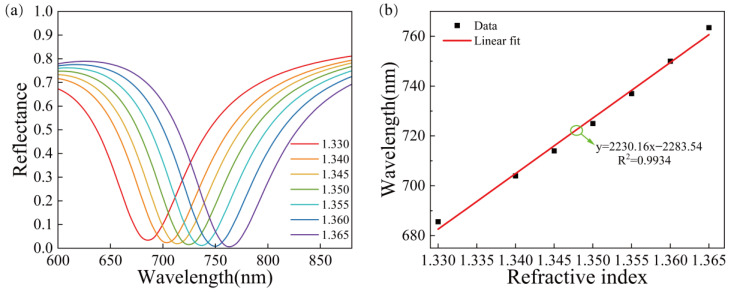
The reflectance curves of the Au/SiO_2_ thin film structure, (**a**) SPR curves for different refractive indices of the Au/SiO_2_ thin film structure, (**b**) relationship between refractive index and resonance wavelength and the linear fitting.

**Figure 6 sensors-23-03163-f006:**
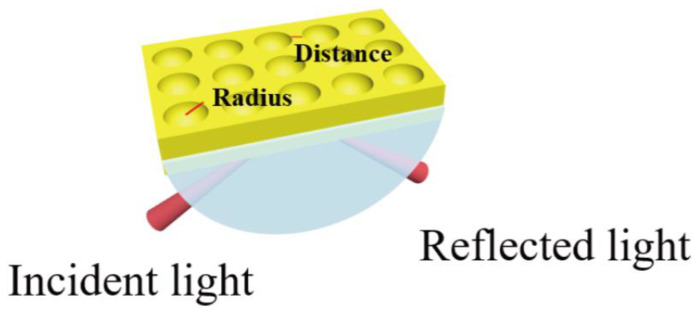
Schematic diagram of the Au array structure SPR sensors.

**Figure 7 sensors-23-03163-f007:**
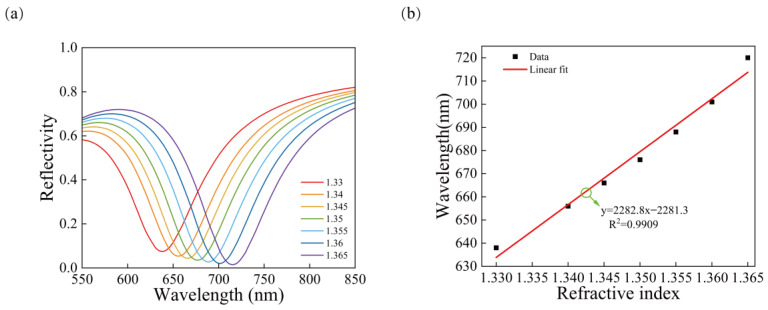
The reflectance curves of the Au array structure, (**a**) SPR curves for different refractive indices of the Au array structure, (**b**) relationship between the refractive index and the resonance wavelength and linear fitting.

**Figure 8 sensors-23-03163-f008:**
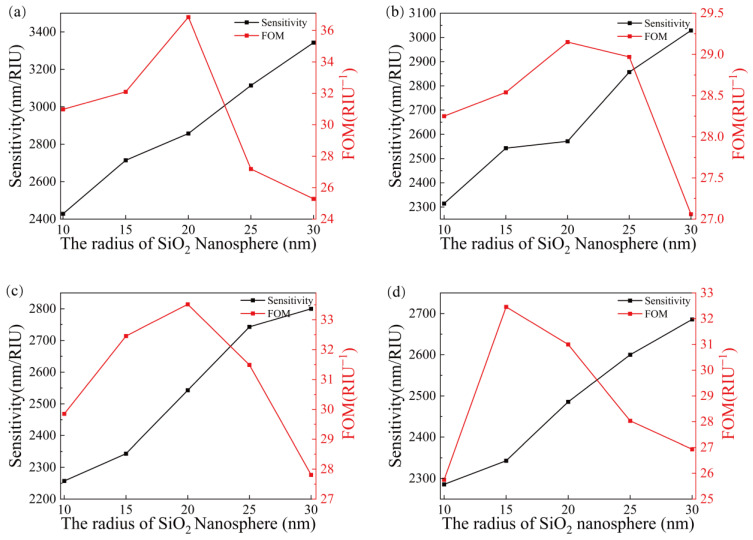
Sensitivity and *FOM* for the different combinations of the nanosphere radius and distance between nanospheres, (**a**) distance of 10 nm, (**b**) distance of 20 nm, (**c**) distance of 30 nm and (**d**) distance of 40 nm.

**Figure 9 sensors-23-03163-f009:**
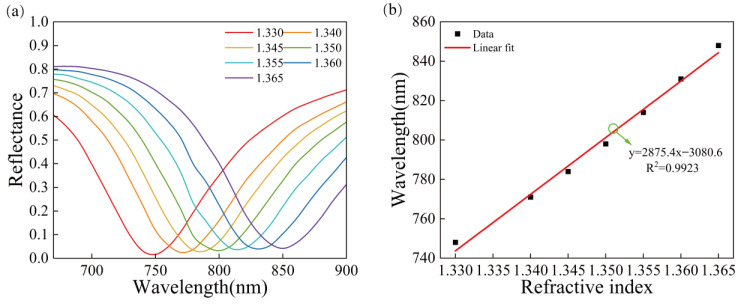
The reflectance curves of the Au/SiO_2_ nanosphere structure, (**a**) SPR curves of the Au/SiO_2_ nanosphere structures with different refractive indices, (**b**) relationship between the refractive index and the resonance wavelength and the corresponding linear fitting.

**Figure 10 sensors-23-03163-f010:**
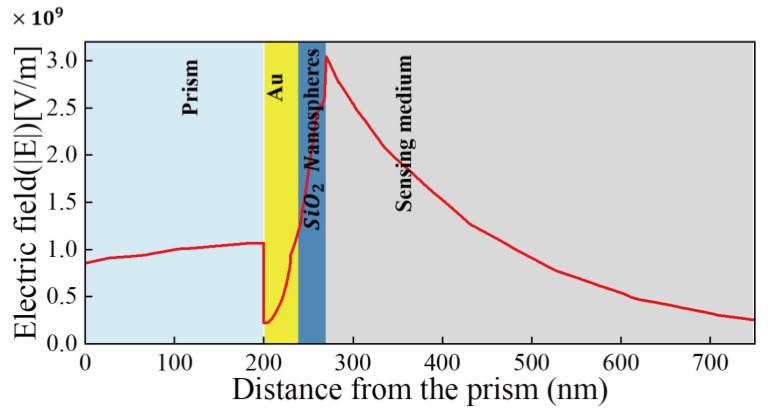
Electric field distribution (the red line) at different interfaces at the resonance wavelength for the medium refractive index 1.330.

**Figure 11 sensors-23-03163-f011:**
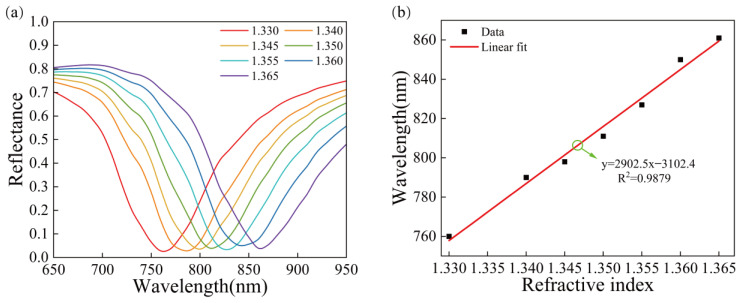
The reflectance curves of the Au/SiO_2_ nanorod structure, (**a**) SPR curves for different refractive indices of the Au/SiO_2_ nanorod structures, (**b**) relationship between the refractive index and the resonance wavelength and linear fitting.

**Figure 12 sensors-23-03163-f012:**
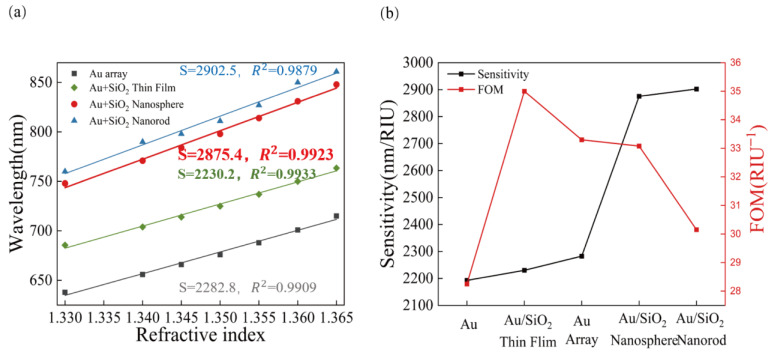
Comparison of the sensitivity of the five structures, (**a**) comparison of sensitivity and (**b**) comparison of the sensitivity and *FOM*.

**Table 1 sensors-23-03163-t001:** Evaluation of the sensing performance for the sensors with different thicknesses and distances between SiO_2_ nanospheres.

Distance (nm)	Radius (nm)	Wavelength (nm) *n* = 1.33	Wavelength (nm) *n* = 1.365	Sensitivity (nm/RIU)	FWHM (nm)	*FOM* (/RIU)
10	20	748	848	2875.4	86.359	33.30
20	25	744	844	2857.14	98.637	28.97
30	20	695	784	2542.9	84.045	30.26
40	20	682	769	2600	92.764	28.03

**Table 2 sensors-23-03163-t002:** Evaluation of the sensing performance for different distances between SiO_2_ nanorods.

Distance (nm)	Wavelength (nm) *n* = 1.33	Wavelength (nm) *n* = 1.365	ΔWavelength (nm)	Sensitivity (nm/RIU)	FHWM (nm)	*FOM* (/RIU)
10	814	927	113	3228.57	119.372	27.05
20	760	861	101	2885.71	95.712	30.15
30	729	824	95	2714.29	90.578	29.97

**Table 3 sensors-23-03163-t003:** Evaluation of the sensing performance for the different radii of the SiO_2_ nanorods.

Radius (nm)	Wavelength (nm) *n* = 1.33	Wavelength (nm) *n* = 1.365	ΔWavelength (nm)	Sensitivity (nm/RIU)	FHWM (nm)	*FOM* (/RIU)
10	711	803	92	2628.57	87.754	29.95
15	740	838	98	2800	93.653	29.90
20	760	861	101	2885.71	95.712	30.15
25	777	881	104	2971.43	103.561	28.69
30	787	893	106	3028.57	107.691	28.12
35	792	897	105	3000	111.620	26.88

**Table 4 sensors-23-03163-t004:** Evaluation of the sensing performance for the different heights of the SiO_2_ nanorods.

Height (nm)	Wavelength (nm) *n* = 1.33	Wavelength (nm) *n* = 1.365	ΔWavelength (nm)	Sensitivity (nm/RIU)	FHWM (nm)	*FOM* (/RIU)
20	700	787	87	2485.7	80.135	31.02
40	758	859	101	2885.71	95.712	30.15
60	760	862	102	2914.29	100.036	29.13

**Table 5 sensors-23-03163-t005:** Performance comparison of the different SPR sensors.

	Sensitivity to the Refractive Index	References
100 nm gold film with 200 nm diameter periodic hole array	400 nm/RIU	[30]
A gold film with an Al_2_O_3_ grating	461.53 nm/RIU	[44]
This work	2875 nm/RIU	

## Data Availability

All the data available in the main text.

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
