# Peer review of "Design and Simulation of Au/SiO2 Nanospheres Based on SPR Refractive Index Sensor"

_sensors, 2023, doi:10.3390/s23063163_

Round 1

Reviewer 1 Report

The authors present a computational investigation of different gold-silica structures for application as SPR sensors. They compare bare gold film with silica coated gold and different silica nano-structures embedded in the gold.

The simulations show that the embedded nanostuctures increase the sensitivity and FOW. However, the authors should discuss what causes the increase and how the structure affects the sensing of molecules in low concentrations. The authors state that:"...the gold film is too smooth to adsorb a large number of molecules, thus limiting the sensitivity ...". Silica nanoparticles are likely not preferential adsorption sites for analytes. The authors should compare the Au hole array that is formed by embedding the silica spheres with and without the silica spheres present. If the enhancement is mainly due to structuring the Au film, I would expect that a holy Au film is preferable since it has a much larger effective surface area than the same film with the silica spheres embedded in the holes. This additional simulation should be quite easy to preform since the SPR structure is already set-up.

Some minor comments:

Page 2 line 53: Citation [33] discusses titania and not silica structures.

I am not sure if Fig. 1 is needed. It’s the same as Fig. 2b.

Page 4 line 120: It is not clear where this result comes from:

"The SPR sensor has the lowest reflectance and the best sensing effect when an Au

film of 50 nm is deposited on ..."

Page 5 line 152: The authors should discuss the error of their method

P2 L53: "SiO2 with" -> "SiO2 structures with"

Author Response

Thank you for your letter and for the reviewers’ comments concerning our manuscript entitled “Design and Simulation of Au/SiO2 Nanospheres Based SPR Refractive Index Sensor” (sensors-2240248). Those comments are valuable and helpful for revising and improving our paper, as well as the important guiding significance to our research. We have studied the comments carefully and have made a correction which we hope meet with approval. Revised portions are marked in red on the paper. The main corrections in the paper and the responses to the reviewer’s comments are as follows:

Reviewer 2 Report

The manuscript entitled "Design and Simulation of Au/SiO2 Nanospheres Based on SPR Refractive Index Sensor" by Sun et al. presented a list of simulation results of Au/SiO2 SPR sensor structure, including plain film, nanorod, nanosphere. They found that nanosphere structure yielded the best performance FOM. While this manuscript performed extensive simulation works of all kinds of structures, however, it worth mentioning that a scientific ARTICLE is NOT a REPORT which merely present what have been done and what the results are. This manuscript does not contains in-depth discussion of the simulation result nor the physical significance of the findings. For example,  the SPR resonance parameter such as FWHM and resonance location are intrinsic properties of a given structure and can be analytically connected to the period, size and loss. Also, silica sphere is the best structure as the conclusion stated, however, how such sphere can be fabricated on a gold film is not considered at all. Therefore, I could not recommend publication of this manuscript in the present state.

Author Response

(The authors gave the same response as above.)

Reviewer 3 Report

The authors designed three different structures of surface plasmon resonance (SPR) sensors by adding three different forms of SiO2 material behind the gold film of the conventional Kretschmann-based sensor to enhance the sensitivity. The problems faced by SPR sensors and some existing solutions are described. Through modeling and simulation, the influence of SiO2 shape on SPR sensors is studied. In summary, this article sounds interesting, but needs to be revised before publication.

1. As a important factor, the gold film thickness should be explained in the corresponding supporting data or supporting literature.

2. Please give possible reasons for the improved sensitivity of different SiO2 morphologies

3. In the conclusion section, the authors summarize the sensitivity enhanced by different forms of SiO2. The significance of this study or the corresponding outlook shoudld be added.

Author Response

(The authors gave the same response as above.)

Round 2

Reviewer 1 Report

The authors addressed my concerns. However, it looks like they did not add the results/conclusion of the gold nano-array without SiO2 (response 1) to the manuscript. I still think a sentence like: "Comparing a pore structure with a SiO2 nanoarray showed that while the structure increased sensitivity above the bare film, the SiO2 nanosphere further increased sensitivity."

I leave it to the authors to decide if they want to include this result.

Author Response

(The authors gave the same response as above.)

Reviewer 2 Report

The authors have properly addressed my concerns. This manuscript can be published after a grammar check.

Author Response

(The authors gave the same response as above.)
